# Mitogenomic Codon Usage Patterns of Superfamily Certhioidea (Aves, Passeriformes): Insights into Asymmetrical Bias and Phylogenetic Implications

**DOI:** 10.3390/ani13010096

**Published:** 2022-12-27

**Authors:** Hengwu Ding, De Bi, Shiyun Han, Ran Yi, Sijia Zhang, Yuanxin Ye, Jinming Gao, Jianke Yang, Xianzhao Kan

**Affiliations:** 1Anhui Provincial Key Laboratory of the Conservation and Exploitation of Biological Resources, College of Life Sciences, Anhui Normal University, Wuhu 241000, China; 2College of Landscape Engineering, Suzhou Polytechnic Institute of Agriculture, Suzhou 215000, China; 3The Institute of Bioinformatics, College of Life Sciences, Anhui Normal University, Wuhu 241000, China

**Keywords:** mitogenome, superfamily certhioidea, phylogeny, codon usage pattern

## Abstract

**Simple Summary:**

The mitochondrial genome (mitogenome) recently has been extensively used in evolutionary analyses. The superfamily Certhioidea is a highly diverse group within the passerine clade, and the phylogeny of this group is still controversial. To date, few studies have focused on the mitogenome evolution of Certhioidea. In the present study, we provided six new complete mitogenomes of Certhioidea. Comprehensive analyses were carried out on the mitogenomes of Certhioidea, including basic genomic characteristics, codon usage patterns, evolutionary rates, and phylogenetic implications. Based on our analyses, we found the codon usage biases of genes were asymmetrical. Most importantly, we suggested that *Salpornis* should be separated from family Certhiidae and put into family Salpornithidae to maintain the monophyly of Certhiidae. The present work may provide new insights on the mitogenome evolution of Certhioidea.

**Abstract:**

The superfamily Certhioidea currently comprises five families. Due to the rapid diversification, the phylogeny of Certhioidea is still controversial. The advent of next generation sequencing provides a unique opportunity for a mitogenome-wide study. Here, we first provided six new complete mitogenomes of Certhioidea (*Certhia americana*, *C. familiaris*, *Salpornis spilonota*, *Cantorchilus leucotis*, *Pheugopedius coraya*, and *Pheugopedius genibarbis*). We further paid attention to the genomic characteristics, codon usages, evolutionary rates, and phylogeny of the Certhioidea mitogenomes. All mitogenomes we analyzed displayed typical ancestral avian gene order with 13 protein-coding genes (PCGs), 22 tRNAs, 2 rRNAs, and one control region (CR). Our study indicated the strand-biased compositional asymmetry might shape codon usage preferences in mitochondrial genes. In addition, natural selection might be the main factor in shaping the codon usages of genes. Additionally, evolutionary rate analyses indicated all mitochondrial genes were under purifying selection. Moreover, *MT-ATP8* and *MT-CO1* were the most rapidly evolving gene and conserved genes, respectively. According to our mitophylogenetic analyses, the monophylies of Troglodytidae and Sittidae were strongly supported. Importantly, we suggest that *Salpornis* should be separated from Certhiidae and put into Salpornithidae to maintain the monophyly of Certhiidae. Our findings are useful for further evolutionary studies within Certhioidea.

## 1. Introduction

The superfamily Certhioidea (5 families, 26 genera, and nearly 150 species) is a highly diverse group within the passerine clade [1,2,3]. It was first erected by Cracraft et al. and initially included four families (Certhiidae, Polioptilidae, Sittidae, and Troglodytidae) that were removed from the superfamily Sylvioidea [4]. Subsequent studies suggested that the wallcreeper (*Tichodroma muraria*) should be placed in the monotypic family Tichodromidae [5].

Over the years, much molecular work has made best efforts to address the phylogeny of superfamily Certhioidea; however, the deep relationships among families are still controversial [1,5,6,7,8,9,10,11,12,13]. To date, a total of four topologies of superfamily Certhioidea had been proposed: (1) the initial topology suggested by Sibley and Ahlquist [6] according to DNA-DNA hybridization and then reconstructed by Zhao et al. [5] based on seven genes (*MT-CYB*, *MT-ND2*, *Myo*, *ODC*, *GAPDH*, *LDH*, and *RAG1*) (Figure 1a); (2) the second topology put forward by Barker [1] derived from six genes (*MT-CYB*, *FGB-I4*, *FGB-I7*, *RAG1*, *RAG2*, and *ZEB1*) (Figure 1b); (3) the third was extracted from the overall phylogenetic tree of Passeriformes by Oliveros et al. [7] on the basis of 4,060 ultra-conserved elements (UCE) in nuclear loci (Figure 1c); and (4) the fourth was inferred by Päckert [8] based upon three mitochondrial genes (*MT-CYB*, *MT-CO1*, *MT-ND2*) (Figure 1d).

In contrast with the complex mitogenomes in plants [14,15,16], metazoan animals generally have compacted mitogenomes [17,18,19,20]. With the advent of next-generation sequencing (NGS), a growing number of avian mitogenomes have been sequenced and analyzed since the first mitogenome of chicken was sequenced [21]. Owing to the small size, high evolutionary conservation, and the absence of introns, the mitogenome has been broadly considered as a valuable tool for avian evolutionary analyses [22,23,24,25]. Up to now, nearly 1000 representative mitogenome sequences of birds are available from GenBank (accessed on 15 November 2022). However, only 16 mitogenomes representing 8 genera in the superfamily Certhioidea have been reported. In a recent mitophylogenetic study, Mackiewicz et al. [22] tried their best to collect as many complete passerine mitogenomes as possible; however, only three sequences from Troglodytidae and one sequence from Sittidae were sampled. Hence, focusing on the phylogeny of superfamily Certhioidea at the mitogenome-wide scale is quite necessary.

As is well known, genetic codes are degenerate and each amino acid residue is encoded by multiple synonymous codons. However, the synonymous codons are not equally utilized, which is referred to as codon usage bias (CUB) [26,27,28]. The CUB is species- and gene-specific and is mainly shaped by the balance force between mutation and natural selection [29,30,31,32]. Therefore, analyzing CUB can be helpful to understand the genomic architectures and evolutionary characteristics of organisms. The CUB analyses were performed in some taxa at mitogenome-wide, such as silkworms [33,34], flies [35], and laughing thrushes [36]. Unfortunately, to our knowledge, little or no scientific work on CUB of mitochondrial genes across Certhioidea taxa has been reported. Thus, investigating the CUB pattern might provide valuable insights in the phylogeny of Certhioidea.

In this study, we successfully assembled and annotated six new complete mitogenome sequences of Certhioidea using the genome skimming approach. Based on our new mitogenome sequences, along with other available sequence data from NCBI database, we tried to address: (1) general characterizations of Certhioidea mitogenomes; (2) codon usage patterns of Certhioidea taxa; (3) rates and patterns of molecular evolution of mitogenomes within Certhioidea; and (4) mitophylogenetic relationships within the Certhioidea.

## 2. Materials and Methods

### 2.1. Data Acquisition, Mitogenome Assembly, and Annotation

In this study, six new mitogenome sequences of superfamily Certhioidea were retrieved from NCBI SRA database by third-party annotation (TPA) (Appendix A). The new data contained four first mitogenomes from three genera (*Salpornis*, *Cantorchilus*, *Pheugopedius*) and two from *Certhia*. The assembly and annotation of mitogenomes were performed using GetOrganelle 1.7.1 [37] and GeSeq [38], respectively. Furthermore, the nomenclature of gene names followed the criterion proposed by HUGO Gene Nomenclature Committee [39].

### 2.2. Nucleotide Compositions, Codon Usage Indices and Evolutionary Rates

Sixteen mitogenome sequences obtained from NCBI, along with six new data points from this study, were used for further analyses. The nucleotide compositions for L-strands or H-strands of whole mitogenome were calculated by using BioEdit 7.2.6 [40].

With exclusion of the termination codons (TAA, TAG, AGG, AGA, and T--), the codon usage indices of PCGs, including the relative synonymous codon usage (RSCU), effective number of codons (ENC), the GC content at codon sites 1 and 2 (GC12), 3 (GC3), and synonymous 3 (GC3s), were measured. The RSCU value for a codon represents the observed frequency divided by that expected (RSCU > 1 and implies a codon used more frequently than expected and vice versa). In detail, RSCU, ENC, and GC3s were estimated by CodonW 1.4.4 [41]. GC12 and GC3 were analyzed by MEGA X 10.0.5 [42]. The percentage of variable sites (PV) and average pairwise nucleotide diversity (π) values were measured with DnaSP v6.12 [43]. The nonsynonymous substitution rate (dN), synonymous substitution rate (dS), and dN/dS ratio were inferred with PAML v4.9 under F3X4 and M0 models (the dN/dS ratio >1, =1, and <1 indicate positive, neutral, and purifying selection, respectively) [44].

### 2.3. Statistical and Graphic Analyses 

Statistical tests and linear regression analyses were computed with OriginPro^®^ 2021 software (OriginLab Corporation, Northampton, MA, USA). The former included mean and standard deviation (SD), while the latter involved slope and intercept. To better display the main codon usage patterns of mitochondrial genes among Certhioidea birds, the parity rule 2-bias (PR2) plots, neutrality plots, and ENC-GC3s plots were drawn with ggplots2 package [45] under R x64 4.0.2 (R Core Team, Vienna, Austria).

### 2.4. Mitophylogenetic Analyses

To better elucidate the evolutionary history of Certhioidea taxa, we conducted mitophylogenetic analyses. Twenty-two Certhioidea species were regarded as ingroups. Additionally, two species from the genus *Regulus* (*R. regulus*, NC_029837; *R. calendula*, NC_024866) were selected as outgroups. All coding regions (13 PCGs without termination codons, 2 rRNAs, and 22 tRNAs) were employed, and multiple sequence alignments were carried out using MAFFT online [46].

The maximum-likelihood (ML) method was performed using RAxML 8.2.12 [47] with 100 ML search runs, 1000 thorough bootstrap replicates, and bootstrapping convergence criterion under the GTRGAMMA model. Prior to the Bayesian inference (BI) analyses, the ModelTest-NG 0.1.6 [48] was used to infer the best-fit models according to Bayesian information criterion (BIC). Subsequently, the BI analyses were conducted by using MrBayes 3.2.7a [49] with two simultaneous runs and four independent MCMC chains (100,000,000 generations, sampling every 10,000th generation). Convergence was checked with the combined effective sample size (ESS) by using Tracer 1.7.1 [50].

## 3. Results and Discussion

### 3.1. General Features of Certhioidea Mitogenomes

#### 3.1.1. Genome Sizes and Gene Contents

Our results provide six new complete mitogenomes of Certhioidea: *C. americana*, *C. familiaris*, *S. spilonota*, *P. coraya*, *P. genibarbis*, and *C. leucotis* (accession numbers: BK016977-BK016982). The accession numbers of the sequences investigated in this study are listed in Table 1. Similar to our previous studies in birds [51,52,53], we detected 37 typical mitochondrial genes from investigated data, including 13 PCGs, 2 rRNAs, and 22 tRNAs, as well as one CR (Figure 2). The sizes of Certhioidea mitogenomes ranged from 16,713 (*Pheugopedius genibarbis*) to 16,920 bp (*Certhia brachydactyla*) (Appendix A). It has been noted that the unusual start codon GTG was observed in *MT-CO1* from 7 Sittidae birds, which might be an apomorphy of the family Sittidae (Appendix A).

In addition, mitogenomes in different avian lineages displayed diversified gene recombination. According to the mitochondrial gene orders, five rearrangement types were proposed in birds: (1) ancestral avian; (2) duplicate CR; (3) remnant CR; (4) duplicate MT-TT–CR; and (5) Hereafter MT-TP–CR [59,60]. In the current study, all investigated mitogenomes of Certhioidea have a typical ancestral avian gene order which was first reported in chicken [21]. The remaining four types could be observed in *Amazona* parrots [61], *Thalassarche albatrosses* [62], *Falco peregrinus* [63], and *Calidris pugnax* [60], respectively. These gene rearrangement events might result from the processes of tandem duplication and random loss (TDRL) [64,65,66].

#### 3.1.2. Asymmetry in Nucleotide Compositions of Certhioidea Mitogenomes

As we know, animal mitochondrial DNA has two strands. Typically, the two mitochondrial DNA strands could be separated by CsCl density gradient centrifugation [67,68]. In this way, an animal mitochondrial genome could be divided into a H-strand (higher G + T content) and L-strand (lower G + T content) [68,69,70]. Thus, the nucleotide composition of the two strands exhibits strand-biased compositional asymmetry (SCA).

In our current study, within 22 mitogenomes of the superfamily Certhioidea, the overall G + T contents of L-strands (38.4% ± 1.0%) are far lower than those of H-strands (61.7% ± 0.9%), which exhibits extreme SCA (Figure 3, Appendix A). This asymmetry of overall nucleotide compositions could be explained based on the strand displacement model of mitochondrial DNA [69]. During the DNA replication process, the spontaneous deamination of A and C could respectively bring I (hypoxanthine) and U mutation in template strands; hence, the newly synthesized strand tends to accumulate more U → C and G → A mutations (I:C pair replaced A:U pair, and U:A pair replaced C:G pair) [71,72,73,74]. Specifically, for most vertebrate mitochondrial genomes, H-strands become rich in G and T, whereas L-strands comprise more A and C.

However, paradoxically, many published studies for vertebrate mitochondrial genomes have described the H-strand as the lower G + T content one with higher amounts of encoding genes, such as humans (GT content = 37.8%) [20], tube-nosed bats (GT content = 42.3%) [75], and Reeve’s turtles (GT content = 39.9%) [67]. These seemingly contradictory results have been possibly explained by Lima and Prosdocimi [68]. They inferred that the phenomena might be caused by the historical reasons that most researchers had neglected the correct assignment of mitochondrial strands since Taanman [20] proposed most vertebrate mitochondrial genes were on the H-strand. Combining the results of this study and previous reports [68,69,70], we proposed that most vertebrate mitochondrial genes should be on the L-strand.

### 3.2. Codon Usage Patterns of PCGs

#### 3.2.1. Asymmetry in Codon Usages of Mitochondrial Genes

The detailed information on RSCU values could be seen in Appendix A. Most surprisingly of all, the RSCU analyses revealed that the over-represented (RSCU_mean_ > 1.6) and preferred (1.0 < RSCU_mean_ ≤ 1.6) codons of all 12 PCGs encoded by L-strands predominantly end with A or C (Appendix A). In fact, some previous studies indicated that ATP genes (*MT-ATP6* and *MT-ATP8*) [76], *MT-CYB* [77], CO genes (*MT-CO1*, *MT-CO2*, and *MT-CO3*) [78], and *MT-ND1* [79] of birds prefer to use A/C-ending codons. Our current study reconfirmed these results among Certhioidea birds. In contrast, different from the codon usage pattern of L-strand genes, the *MT-ND6* encoded by H-strands preferred to use the G/T-ending codons (Appendix A and Appendix A).

Furthermore, the GC bias [G3/(G3 + C3)] and AT bias [A3/(A3 + T3)] of each gene among 22 birds were analysed. According to PR2 plot, the points were at the center (GC bias = 0.5, AT bias = 0.5), meaning no bias for mutation and selection; however, the off-centered points reflect the existence of bias [80]. In our analysis, the uniform PR2 pattern (points almost lied on the second quadrant with GC bias < 0.5 and AT bias > 0.5) was observed among all L-strand genes (Figure 4). Most interestingly, the reversed PR2 pattern (points lied on the fourth quadrant with GC bias > 0.5 and AT bias < 0.5) was detected in *MT-ND6* (Figure 4). These off-centered distributions confirmed that the role of mutation pressure and natural selection in shaping the codon usage of mitochondrial genes among Certhioidea birds. In addition, these results indicated again that the H-strand genes and L-strand genes are inclined to employ G/T-ending and A/C-ending codons, respectively. Our study indicated that the SCA might shape the codon usage pattern of both H-strand and L-strand genes in avian mitogenomes. Notably, we found that two points representing *C. brachydactyla* and *C. familiaris* were markedly different from other points in the PR2 plot of *MT-ATP8* compared with other *Certhia* species. As reported by Abdoli et al. [33], similar asymmetric distributions of PR2 plots were observed in mitochondrial genes of silkworms. In that work, four H-strand genes and nine L-strand genes obviously preferred to use G/T-ending codons and C-ending codons (the preferences of A-ending codons were not obvious), respectively [33]. In contrast, due to the differences in gene numbers in H-strands and L-strands, the PR2 bias patterns of Certhioidea birds were quite different from those of silkworms.

To our knowledge, the asymmetrical codon usage patterns of PCGs were first found in the mitogenome of *Asterina pectinifera* (echinoderm), where three mitochondrial genes encoded by H-strands tend to use G/T-ending codons, whereas the remaining 10 genes on the opposite strand prefer to use A/C-ending codons [81]. After that, the asymmetrical codon usage patterns were found in some vertebrate mitogenomes, such as mammalian mitogenomes, suggesting that L-strand genes are fond of A/C-ending codons [82]. In addition, similar phenomena were also be found in several bacterial genomes, such as *Borrelia burgdorferi* [83], *Tropheryma whipplei* [84] and *Lawsonia intracellularis* [85], that the genes encoded by leading strand favored G/T-ending codons, whereas those on the lagging strand preferred to use A/C-ending codons. Most interestingly, these phenomena were only found in circular genomes.

#### 3.2.2. Main Driving Factor in Codon Usages of Mitochondrial Genes

The ENC and GC3s values of 13 genes among Certhioidea birds were calculated by CodonW. Consequently, the analysis results showed huge variations among genes and species, with the ENC values ranging from 27.85 (*MT-ND4L* in *Sitta nagaensis*) to 50.38 (*MT-ND1* in *Certhia familiaris*) (Appendix A), and the GC3s values extending from 0.327 (*MT-ATP8* in *Polioptila caerulea*) to 0.704 (*MT-ND4L* in *C. familiaris*) (Appendix A). Generally, the ENC value ≤ 35 indicates significant codon usage bias (CUB) [86,87]. Among a total of 286 sequences (13 genes from 22 species), 54 (18.88%) exhibited significant CUB, while the remaining 232 (81.12%) exhibited relatively weak CUB.

ENC-GC3s plot analysis is an efficient tool for verifying the main driving factor of codon usage bias (mutational bias or natural selection) [36,88,89]. If the codon usage is only influenced by mutational bias, the points are expected to lie on or just below the expected ENC curve; alternatively, if one gene is under pure natural selection, it is far away from the expected curve [36,88,89]. As can be seen from Appendix A, the points of all 13 PCGs from Certhioidea mitogenomes are distributed well below the curve, illuminating the predominance of natural selection pressure over mutational bias. Interestingly, the ENC-GC3s point distributions of 13 PCGs presented in this report were found highly similar to those of Laughing thrushes [36].

Neutrality plots of the 13 PCGs were drawn to estimate the extent of influence of mutation pressure and natural selection on the CUB. The regression coefficient (slope) of neutrality plot is regarded as the mutation–selection equilibrium frequency. Here, a slope of 1 represents complete neutrality, while 0 shows complete selective constraint [90,91]. Furthermore, a slope of less than 0.5 could reflect that selection might have played a major role in shaping CUB, with greater than 0.5 and less than 1 indicating dominant influence of mutation pressure [33]. Previous studies indicated the CUB of *MT-ATP8* (0.063) [76] and *MT-CYB* (0.024) [92] of birds were mainly shaped by selection. In this study, as shown in Figure 5, the absolute values of the slopes of 13 PCGs ranged from 0.0052 (*MT-ND6*) to 0.1364 (*MT-ND5*). All these values were much lower than 0.5, suggesting that natural selection played a more important role than mutation in shaping the CUB of mitochondrial genes among Certhioidea taxa. In addition, these regression coefficients showed that the contributions of mutation pressure were 0.52%, 1.68%, 2.53%, 4.30%, 5.08%, 5.47%, 7.10%, 8.18%, 8.22%, 11.13%, 11.49%, 12.24%, and 13.64% for *MT-ND6*, *MT-CO1*, *MT-CYB*, *MT-ND1*, *MT-ATP8*, *MT-ND4L*, *MT-ND4*, *MT-ND3*, *MT-CO3*, *MT-ATP6*, *MT-CO2*, *MT-ND2*, and *MT-ND5*, respectively.

As we know, natural selection on codon usage can increase translation accuracy and efficiency, resulting in a reduction in global translation costs [93]. Through comprehensive analyses of ENC-GC3s and neutrality plots, we demonstrate the major role of natural selection in evolution of codon usage for Certhioidea mitogenomes. In order to better investigate the CUB of mitochondrial genes among birds, more endeavors are needed for further studies.

### 3.3. Evolutionary Rates and Patterns

To better understand the evolutionary patterns of mitochondrial PCGs, some indices, including PV, π, dN, dS, and dN/dS values, were evaluated within the superfamily Certhioidea. Among the 13 mitochondrial PCGs, *MT-ATP8* is the most divergent gene by PV value (53.94%), followed by *MT-ND2* (52.31%), and *MT-ND6* (51.55%). By contrast, the lowest two are *MT-CO1* (36.05%) and *MT-CO3* (39.21%). The π values of PCGs ranged from 0.13119 to 0.18695 (Table 2).

The dN values varied between 0.073 and 1.0157. Surprisingly, the dS values displayed relatively wide ranges (5.7834–19.4011) compared to dN values, which resulted in significantly lower dN/dS ratios (0.00604–0.17562) (Table 2). These results indicated the mitochondrial PCGs of Certhioidea species appear to be evolving under purifying selection. In this study, the *MT-ATP8* (dN/dS = 0.17562) and *MT-CO1* (dN/dS = 0.00604), respectively, are the most rapidly evolving gene and most conserved gene among the 13 PCGs, which is congruent with our previous studies in Accipitriformes [94], Passeriformes [51], and Piciformes [95].

Mitochondrial genes play a pivotal role in the process of oxidative phosphorylation [96,97,98]. Nonsynonymous substitutions are generally harmful in respiratory-chain activity, which might limit the energy biosynthesis [99,100]. To maintain functional requirements, most genes (dN/dS < 0.1), especially for *MT-CO1*, experienced strong evolutionary constraints [101,102], whereas *MT-ATP8* (dN/dS > 0.1) has evolved more quickly than others. It is interesting why *MT-ATP8* evolved so fast. The probable reason is that fixation of advantageous mutations in *MT-ATP8* might result in accelerated evolution of other mitochondrial genes by compensation-draft feedback (CDF) process [103,104]. Therefore, these findings confirm that *MT-ATP8* might play important roles in the evolution of avian mitogenomes.

### 3.4. Phylogenetic Implications

Currently, there is a paucity of research focusing on the phylogeny of superfamily Certhioidea at the mitogenome-wide level. Our study sampled all the families of superfamily Certhioidea to reconstruct the mitophylogenetic tree. The best-fit models of partitioned analyses can be seen in Appendix A. The bootstrap convergence assessment checked by RAxML showed that the ML analysis converged after 150 replicates. The ESS checked by Tracer was 2843.7 (>200), indicating the BI analyses were also convincible. The trees derived from ML and BI methods displayed the same topology (Figure 6). The mitophylogenetic tree inferred by our study is similar with the initial topology proposed by Sibley and Ahlquist [6].

Troglodytidae is monophyletic and sister to Polioptilidae (represented by *P. caerulea*) with high supports (BS = 100 and PP = 1.00), which agrees with many previous reports [1,5,7,8,13,105]. We further divided Troglodytidae into two distinct clades (clade A and clade B). Within clade A, two species of the genus *Pheugopedius* (*P. coraya* and *P. genibarbis*) are sister to (*H. leucosticte* and *C. leucotis*) (BS = 100 and PP = 1.00). Meanwhile, within clade B, *T. ludovicianus* and two *Campylorhynchus* species (*C. zonatus* and *C. brunneicapillus*) were generally identified as sister taxa (BS = 65 and PP = 1.00), with *T. troglodytes mosukei* in a basal position. Notably, three species (*C. leucotis*, *P. coraya*, *P. genibarbis*) in clade A were formerly treated as members of *Thryothorus* but have been classified into two novel genera (*Cantorchilus* and *Pheugopedius*) by Mann et al. [106]. As is seen in Figure 6, *Henicorhina leucosticte* is embedded in these three species. Therefore, our results support the above taxonomic revisions.

In addition, the genus *Salpornis* (represented by *S. spilonota*) might have a relatively closer relationship to the genus *Certhia* with much lower support values (BS = 33 and PP = 0.59). According to the Clements Checklist 2022 [2] and IOC World Bird List v. 12.1 [3], these two genera (*Certhia* and *Salpornis*) make up the current family Certhiidae. However, Oliveros et al. [7] put *Salpornis* into a novel family Salpornithidae to maintain the monophyly of Certhiidae (Figure 1c). Moreover, in contrast with the study of Zhao et al. (Figure 1a) [5], obviously decreased support values at the node of (*Salpornis* and *Certhia*) were observed in our study (BS: 51 → 33; PP: 0.99 → 0.59). Therefore, in order to keep the monophyly of Certhiidae, we also suggest *Salpornis* should be classified into the Salpornithidae family. Furthermore, Sittidae, including seven *Sitta* species, is sister to Tichodromidae (represented by *T. muraria*) with a low BS value but a high PP value (BS = 48 and PP = 1.00) in our study. These results were similar to those of Zhao et al. (Figure 1a) [5].

According to our analyses, the monophylies of Troglodytidae and Sittidae were strongly supported. However, there are still many unsolved phylogenetic problems within superfamily Certhioidea, especially the relationships among families of Certhioidea, which are not entirely clear. In order to better understand the phylogeny of these taxa, more data are needed for the further detailed analyses.

## 4. Conclusions

In the present study, six new mitogenomes of superfamily Certhioidea were reported. The sizes of mitogenomes within Certhioidea ranged from 16,713 to 16,920 bp. It has been noted that the unusual start codon GTG was observed in *MT-CO1* from seven Sittidae birds, which might be an apomorphy of the family Sittidae. Within mitogenomes of Certhioidea, the overall G + T contents of L-strands (38.4% ± 1.0%) are far lower than those of H-strands (61.7% ± 0.9%). Additionally, RSCU values and PR2 plots analyses indicated that H-strand genes preferred to use G/T-ending codons, while L-strand genes tend to utilize A/C-ending codons. These differences might be caused by strand-biased compositional asymmetry. The ENC-GC3s plot and neutrality plot analyses illustrated that natural selection might play a critical role on shaping the codon usages of mitochondrial genes. Furthermore, the regression coefficients of neutrality plots indicated that the effect degrees of mutation pressure ranged from 0.52% to 13.64% for 13 mitochondrial PCGs within Certhioidea. Evolutionary rate analyses indicated all mitochondrial genes of Certhioidea were under purifying selection. Furthermore, *MT-ATP8* (dN/dS = 0.17562) and *MT-CO1* (dN/dS = 0.00604) were the most rapidly evolving gene and conserved gene, respectively. According to our mitophylogenetic analyses, the monophylies of Troglodytidae and Sittidae were strongly supported. Importantly, we suggest that *Salpornis* should be separated from Certhiidae and put into Salpornithidae to maintain the monophyly of Certhiidae.

## Figures and Tables

**Figure 1 animals-13-00096-f001:**
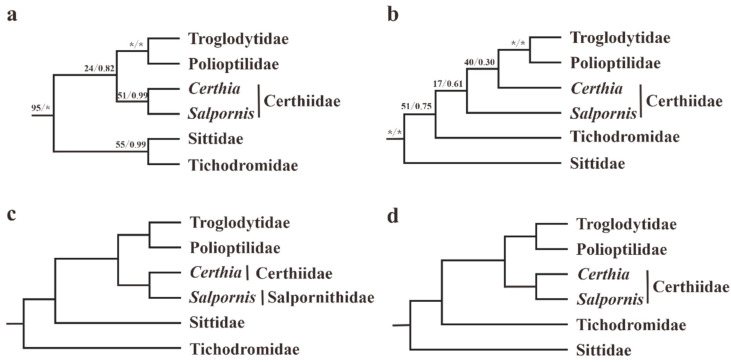
Four representative recently published topologies of superfamily Certhioidea. They were derived from: (**a**) Sibley and Ahlquist (1990), and Zhao et al. (2016); (**b**) Barker (2017); (**c**) Oliveros et al. (2019); (**d**) Päckert et al. (2020). The values at nodes are bootstrap percentage (BP) calculated in RAxML and Bayesian posterior probability (PP) inferred by MrBayes; “*” indicates 100% BS or 1.00 PP.

**Figure 2 animals-13-00096-f002:**
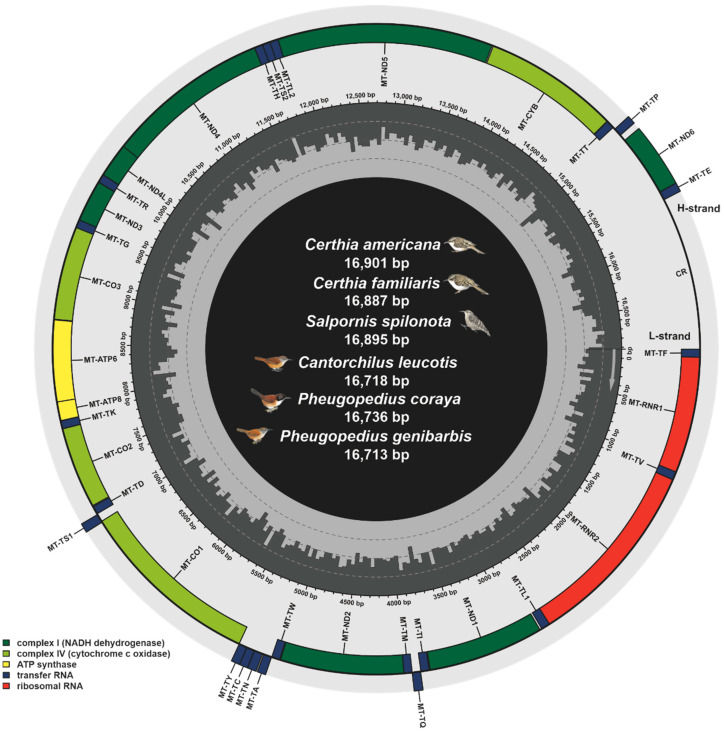
Mitogenome map of 6 Certhioidea species we determined.

**Figure 3 animals-13-00096-f003:**
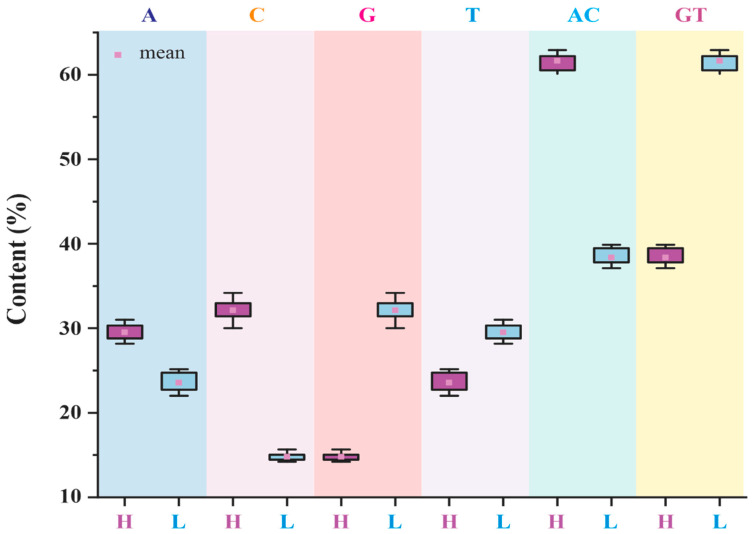
Nucleotide compositions of whole mitogenomes among superfamily Certhioidea species. H and L indicate H-strand and L-stand of mitogenome, respectively.

**Figure 4 animals-13-00096-f004:**
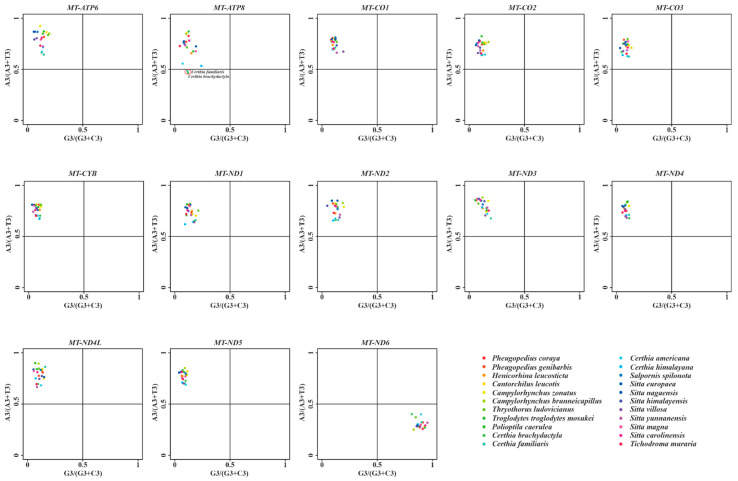
PR2 plots of mitochondrial genes among superfamily Certhioidea species. The *x*-axis: GC-bias [G3/(G3 + C3)]. The *y*-axis: AT-bias [A3/(A3 + T3)].

**Figure 5 animals-13-00096-f005:**
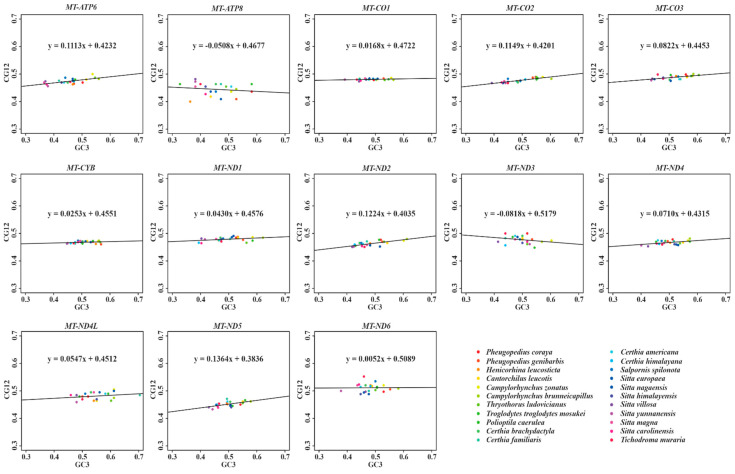
Neutrality plots of mitochondrial genes among superfamily Certhioidea species.

**Figure 6 animals-13-00096-f006:**
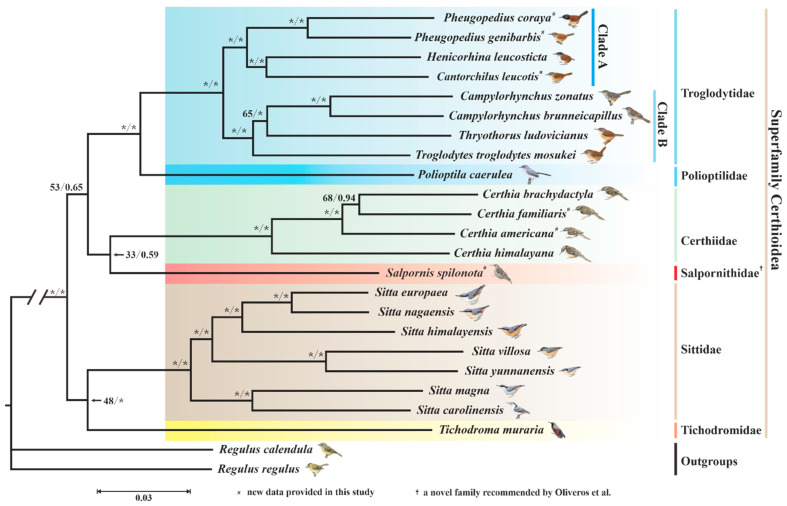
Phylogenetic tree of superfamily Certhioidea. The BS and PP values for each node are indicated; “*” indicates 100% BS or 1.00 PP. ^†^ a novel family recommended by Oliveros et al. [7].

**Table 1 animals-13-00096-t001:** Species of mitogenomes examined in this study.

Family	Species	Accession No.	Reference
Certhiidae	*Certhia americana* (Brown Creeper)	BK016977	This study
Certhiidae	*Certhia brachydactyla* (Short-toed Treecreeper)	NC_053055	[54]
Certhiidae	*Certhia familiaris* (Eurasian Treecreeper)	BK016978	This study
Certhiidae	*Certhia himalayana* (Bar-tailed Treecreeper)	NC_053710	Duan et al. (2018) ^a^
Certhiidae	*Salpornis spilonota* (Indian Spotted Creeper)	BK016979	This study
Polioptilidae	*Polioptila caerulea* (Blue-grey Gnatcatcher)	NC_051031	[54]
Sittidae	*Sitta carolinensis* (White-breasted Nuthatch)	NC_024870	[55]
Sittidae	*Sitta europaea* (Wood Nuthatch)	NC_053059	[54]
Sittidae	*Sitta himalayensis* (White-tailed Nuthatch)	NC_042730	Duan et al. (2018) ^a^
Sittidae	*Sitta magna* (Giant Nuthatch)	MZ888773	Yuan et al. (2021) ^a^
Sittidae	*Sitta nagaensis* (Chestnut-vented Nuthatch)	NC_042731	Duan et al. (2018) ^a^
Sittidae	*Sitta villosa* (Snowy-browed Nuthatch)	NC_051513	[56]
Sittidae	*Sitta yunnanensis* (Yunnan Nuthatch)	MN052793	Duan et al. (2021) ^a^
Tichodromidae	*Tichodroma muraria* (Wallcreeper)	NC_053081	[54]
Troglodytidae	*Campylorhynchus brunneicapillus* (Cactus Wren)	NC_029482	Zhao (2016) ^a^
Troglodytidae	*Campylorhynchus zonatus* (Band-backed Wren)	NC_022840	[57]
Troglodytidae	*Cantorchilus leucotis* (Buff-breasted Wren)	BK016982	This study
Troglodytidae	*Henicorhina leucosticta* (White-breasted Wood Wren)	NC_024673	[58]
Troglodytidae	*Pheugopedius coraya* (Coraya Wren)	BK016980	This study
Troglodytidae	*Pheugopedius genibarbis* (Moustached Wren)	BK016981	This study
Troglodytidae	*Thryothorus ludovicianus* (Carolina Wren)	NC_051032	[54]
Troglodytidae	*Troglodytes mosukei* (Eurasian Wren)	LC541429	Yamamoto et al. (2020) ^a^

Note: ^a.^ These data were directly submitted to NCBI nucleotide database.

**Table 2 animals-13-00096-t002:** Evolutionary rates of mitochondrial PCGs of Certhioidea species.

Gene	Length (bp)	PV (%)	π	dN	dS	dN/dS
*MT-ATP6*	681	48.75	0.17522	0.4205	18.2467	0.02304
*MT-ATP8*	165	53.94	0.17885	1.0157	5.7834	0.17562
*MT-CO1*	1548	36.05	0.13119	0.073	12.0922	0.00604
*MT-CO2*	681	42.29	0.15738	0.2975	11.3325	0.02625
*MT-CO3*	783	39.21	0.14331	0.213	14.1762	0.01503
*MT-CYB*	1140	42.28	0.14973	0.3024	12.8224	0.02358
*MT-ND1*	975	47.69	0.18030	0.3747	17.8404	0.021
*MT-ND2*	1038	52.31	0.18656	0.5995	15.1289	0.03963
*MT-ND3*	348	49.14	0.17068	0.532	14.9662	0.03555
*MT-ND4*	1377	47.79	0.17211	0.4603	19.1301	0.02406
*MT-ND4L*	294	47.62	0.16185	0.2867	19.4011	0.01478
*MT-ND5*	1815	48.15	0.16907	0.4818	16.3115	0.02954
*MT-ND6*	516	51.55	0.18695	0.6426	10.4749	0.06135

## Data Availability

The sequence data generated in this study are available in GenBank of the National Center for Biotechnology Information (NCBI) under the access numbers: BK016977–BK016982.

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
