# Peer review of "Mitogenomic Codon Usage Patterns of Superfamily Certhioidea (Aves, Passeriformes): Insights into Asymmetrical Bias and Phylogenetic Implications"

_animals, 2022, doi:10.3390/ani13010096_

Round 1

Reviewer 1 Report

The article is called "Mitogenomic Codon Usage Patterns of Superfamily Certhioidea (Aves, Passeriformes): Insights into Asymmetrical Bias and Phylogenetic Implications" the investigation into the collection of mitegonome from six species of the Certhioidea family and the identification of their phylogenetic relationships to those species found in the Genbank. In order to ascertain the phylogenetic relationship of the family, the researchers was successful in extracting the mitegonomes from the six species and comparing them to the species in the Genbank.

The article explores the various methods used to analyze bird phylogenetic relationships and dives into the challenges presented by the complexity of the data but I noticed that in the method section, where the samples were obtained from, the previous method sections in the analysis were not included. I think it would be beneficial to include the DNA Extraction and Sequencins sections. This would provide additional context to better understand the sample acquisition process.

The genus Salpornis and its connection to the Certhiidae family are the main ideas of the ms. According to the research, Salpornis belongs in the Salpornithidae family rather than the Certhiidae family. Add more species to help us better understand the families. The inclusion of only 22 species in the manuscript restricts the pylogenetic portion of the article, despite the fact that this superfamily Certhioidea contains 150 species. I think that by increasing the number of species included, the phylogenetic analysis's accuracy might be significantly increased. It would also give a deeper understanding of the variety and evolution of the superfamily Certhioidea.

Some specific comments

Fig 1 (L58 L59) remove the added extras "a" and "Tichodromidae" please also check the other figures too

L94 delete or rephrase the sentence

L97-98 rephrase

L118 gice a space "=1"

L151-153 Also there is in the MT-ND5 in sublementary file too. Please clarify it

Fig 4 title of the x and y axes are not clearly understandble

Fig 5 it should be given as sublementary file

L264 delete ")"

Fig 6 remove the corelation anlysis in the figure. It is meaningless. scale of the x and y axisis should be extended as only show the data range to clearly understand the species locations

L313 delete ">" one of the two

L324 and L338 change the "+" as and

L 366 delete or rephrase the sentence

Reviewer 2 Report

The study presented by the authors is certainly of weighty significance for the disclosure of questions of the phylogenetics of birds and in particular the taxonomic structure of the passerine clade. This is achieved by the fact that the authors, having looked inside this clade, found the need to study the structure of one of its fundamental groups, Certhioidea, and indeed, having investigated the mitogenomics of this group, found that the phylogenetic tree of Certhioidea has a slightly different structure than previously assumed. In doing so, the authors have contributed to the currently relevant idea of mitogenomic evolution by stressing that this phenomenon is peculiar to the bird clade of the passerine.

The relevance and novelty of the study is also highlighted by its methodology. To make their discovery, which did not occur in the previous literature, the authors resort to modern methods such as mitogenome sequencing, for which NGS was applied, not used for the study of these taxa before.

It should also be noted how thoroughly the authors have reviewed previous studies in this field. However, we would like to draw attention to some fundamental aspects to which the authors appeal in their conclusions.

In the Conclusion, you highlighted how your findings demonstrate the importance of the role of natural selection on shaping the codon usages of mitochondrial genes. We have noticed how in previous sections such as the Introduction and Discussion, you argue for natural selection inseparably with reference to mutations (using terms such as mutation pressure and mutation deviation), as if to separate them as an extraneous phenomenon, despite the fact that mutations are one of the driving forces of natural selection and thus natural selection a priori implies them. Could you please explain why you do not then mention mutations in the context of natural selection in the Conclusion?

Reviewer 3 Report

The work is very interesting and extensively described but sometimes the presented results are a bit not clear. The authors should introduce small corrections in this part. 
